

# A Distributed Temperature Sensing based soil temperature profiler

Bart Schilperoort[1,2], César Jiménez Rodríguez[1,3], Bas van de Wiel[4], and Miriam Coenders-Gerrits[1]

[1]Delft University of Technology, Water Management department, Stevinweg 1, 2628 CN Delft, the Netherlands
[2]Netherlands eScience Center, Science Park 402, 1098 XH Amsterdam, the Netherlands
[3]Luxembourg Institute of Science and Technology, Av. des Hauts-Fourneaux, 4362 Esch-sur-Alzette, Luxembourg
[4]Delft University of Technology, Geoscience and Remote Sensing department, Stevinweg 1, 2628 CN Delft, the Netherlands

**Correspondence:** Bart Schilperoort (b.schilperoort@gmail.com)

**Abstract.** Storage of heat in the soil is one of the main components of the energy balance, and is essential in studying the land-atmosphere heat exchange. However, its measurement proves to be difficult, due to (vertical) soil heterogeneity and sensors easily disturbing the soil.

Improvements in precision and resolution of Distributed Temperature Sensing (DTS) equipment has resulted in widespread
use in geoscientific studies. Multiple studies have shown the added value of spatially distributed measurements of soil temperature and soil heat flux. However, due to the spatial resolution of DTS measurements (~30 cm), soil temperature measurements with DTS have generally been restricted to (horizontal) spatially distributed measurements. In this paper a device is presented which allows high resolution measurements of (vertical) soil temperature profiles, by making use of a 3D printed screw-like structure.

A 50 cm tall probe is created from segments manufactured with fused filament 3D printing, and has a helical groove to guide and protect a fiber optic cable. This configuration increases the effective DTS measurement resolution, and will inhibit preferential flow along the probe. The probe was tested in the field, where the results were in agreement with the reference sensors. The high vertical resolution of the DTS-measured soil temperature allowed determination of the thermal diffusivity of the soil at a resolution of 2.5 cm, many times better than feasible with discrete probes.

Future improvements in the design could be integrated reference temperature probes, which would remove the need for DTS calibration baths. This could, in turn, support making the probes 'plug and play' of the shelf instruments, without the need to splice cables or experience in DTS-setup design. The design can also support integrating an electrical conductor into the probe, and allow heat tracer experiments to derive both the heat capacity and thermal conductivity over depth at high resolution.

## 1 Introduction

The exchange of heat between the atmosphere and the land surface is one of the main components of the local energy balance. This heat exchange takes place at the surface, but is driven by the temperature gradient between the surface and soil deeper down. The process is strongly affected by soil cover (vegetation), soil type and the hydraulic and thermal properties of the soil.

In order to study land-atmosphere heat exchange, knowledge of the surface-skin temperature is important (Holtslag and De Bruin, 1988; Heusinkveld et al., 2004). Unfortunately, from an observational perspective, measuring this 'skin' temperature is



very challenging (Van de Wiel et al., 2003). With traditional sensors the upper soil is easily disturbed, as the observation should be done as close to the surface as possible. Moreover, the soil near the surface is strongly heterogeneous due to larger organic matter content as compared to deeper soil layers. As measurements of the 'skin temperature' are made at finite depth, it can thus be questioned how representative these are.

Alternatively, modeling approaches can be followed in order to infer the skin temperature from deeper soil temperatures: as 30 the surface temperature varies with the diurnal rhythm, many analyses focus on the amplitude damping (van Wijk and de Vries, 1963; Van de Wiel et al., 2003), phase shifts, or harmonics (Verhoef, 2004; Heusinkveld et al., 2004; van der Tol, 2012; van der Linden et al., 2021) to model the propagation of heat through the soil. With these methods, the soil temperature and heat flux measured at certain depths are interpolated and extrapolated to infer an entire profile, or the heat flux and temperature at the surface. However, these methods are sensitive to the parameterization of, among others, 'how easily heat moves through the 35 soil', i.e. the soil thermal diffusivity (Xie et al., 2019). For determining the soil thermal diffusivity, the soil temperature at least three depths is required (although if one assumes the soil to be homogeneous over depth, two can suffice). This means that high resolution profiles of thermal diffusivity require an even higher density of temperature measurements. Besides, the vertical variability of organic content and water content will make the diffusivity height dependent.

Not only are the soil temperature models sensitive to the parameterization, great care has to be taken in the placement of the 40 sensors themselves. Near the surface temperatures can be very heterogeneous, due to differences in soil cover or vegetation height. Deeper down this is less of an issue, as any surface differences will smooth out due to lateral diffusion (Eppelbaum et al., 2014). Note that small changes in sensor depth estimates may result in large temperature changes, particularly near the surfaces where gradients are expected to be large. Thus, exact determination of the depth at which the sensors are located is very important as uncertainties in this will propagate through the analysis (Dong et al., 2016). Some soil sensors already take 45 this into account, by affixing multiple temperature sensors to a solid structure which is placed into the soil, ensuring that the relative spacing is accurate down to the millimeter.

In recent years distributed temperature sensing (DTS) has become more prominent in studies of soil temperature and properties. Many of these studies have aimed to measure the spatial distribution of soil moisture, either with passive measurements combined with the soil properties (Steele-Dunne et al., 2010), or by actively heating the fiber-optic cable to gain more infor- 50 mation on soil thermodynamic properties (Sayde et al., 2010; Shehata et al., 2020; Wu et al., 2021) Other studies focused on measuring the spatial distribution of the soil heat fluxes or surface heat flux (Jansen et al., 2011; Dong et al., 2016; Bense et al., 2016). In nearly all these studies the fiber-optic cables were placed *horizontally* in the soil, sometimes with a specially designed plow. Even then, horizontal cable placements will have some uncertainty and small errors in the placement depth strongly affect results (Steele-Dunne et al., 2010). However, simply placing a cable vertically has no use due to the spatial 55 resolution of DTS measurements, which is 0.25 m at its best. To overcome this, the cable can be placed in a *fixed coil* shape (Hilgersom et al., 2016; Saito et al., 2018; Schilperoort et al., 2020; Wu et al., 2020). Affixing the cable to a coil effectively increases the vertical resolution, and can ensure that the distance between each measurement point is both fixed and more accurate than with separate sensors or cables. In Saito et al. (2018), the soil temperature profile in both the top layer of the soil,





and snow covering the soil, was measured using fiber-optic cable wrapped around a large-diameter PCV tube. However, these
large diameter PVC tubes can be challenging to install without disturbing the soil to a great amount.

To this end we designed a DTS-based soil temperature probe that can be placed into an hand auger-dug hole in the soil, using a fiber-optic cable as the screw-thread. In this technical note we discuss the design of the probe, how to build it, and we test the probe in the field with a comparison to reference sensors. We present the temperature profiles and derived diffusivity profiles that can be measured using the probe, as compared to standard discrete sensors. Lastly, we will discuss the limitations
of the design and give an outlook to improvements and future use cases.

## 2 Materials and methods

### 2.1 Probe design

The concept behind the design is twofold; place more optical fiber in a smaller space for a higher resolution, and create a helical screw thread. When the fiber optic (FO) cable that is used for the probe has a large diameter (e.g., 6 mm), the cable *itself* can
act like the screw thread to ensure good contact with the soil. This is required to get a representative temperature measurement, and will also prevent water from flowing straight down along the tube. Such a probe could be constructed by 3D printing or adding a spiral groove in, e.g., a PVC pipe with a lathe. Besides good soil contact, a cable with a low heat capacity and low thermal conductivity is also required to avoid disturbing the temperature profile of the soil. As cables with a large diameter often contain metal, which is highly conductive, we chose to use a thin fiber optic cable as an alternative. However, with smaller
diameter cables, the 'screw thread' would not protrude out from the core as far. This could cause insufficient contact with the soil. A second issue is that smaller diameter cables have less protection for the fiber, which increases the chance of damage during installation.

To mitigate these problems, a protruding screw thread is incorporated into the design (Fig. 1). This creates a screw that makes good contact with the soil, and provides a groove to install the fiber optic cable into.
Inside the probe is an empty central tube. This space runs vertically through the probe, and allows the routing of the fiber optic cable back to the top of the probe. During assembly it is filled in with expanding polyurethane foam to prevent heat transport and to seal out moisture. Hexagonal protrusions and slots are present on the top and bottom of the segments to make alignment easier during assembly. The diameter of the probe (75 mm, excluding the protrusions) was chosen to fit with our available augers, to ensure compatibility between the dug hole and the probe.
For installation, a tool was designed. The tool engages into the holes at the top of the probe (Fig. 1 and has handles to allow screwing the probe into a pre-drilled hole. The tool can be seen in Figure 3. After installation, the three holes are filled with printed plastic cylinders.



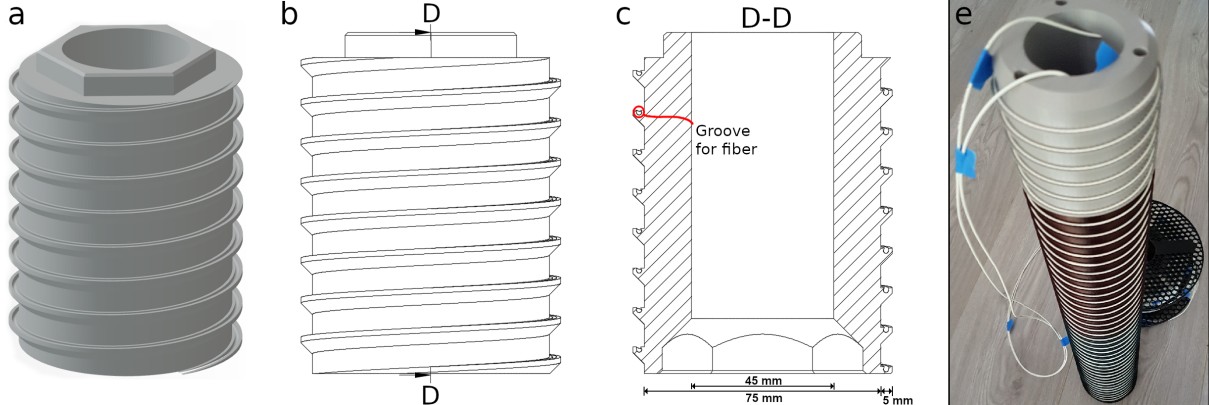

**Figure 1.** 3D render (a), a side view (b) and a vertical cross-section (c) of a segment of the DTS probe. A photo of the probe with all elements assembled is shown in (e).

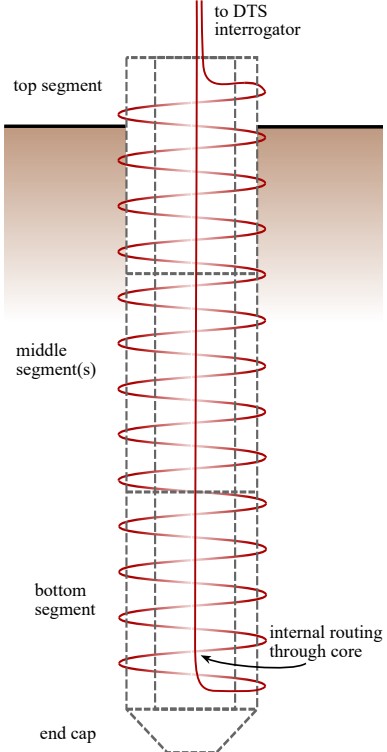

**Figure 2.** Schematic drawing illustrating an installed probe, with a top section, single middle section and bottom section. The fiber routing is illustrated in red.





| Material | (Bulk) density | Specific heat capacity | Volumetric heat capacity | Thermal conductivity |
|---|---|---|---|---|
| | (kg m$^{-3}$) | (J kg$^{-1}$ K$^{-1}$) | ($10^3$ J m$^{-3}$ K$^{-1}$) | (W m$^{-1}$ K$^{-1}$) |
| **PLA** | 1240[a] | 1590[b] | 1971 | 0.11 [b] |
| **PETG** | 1270[a] | 1300[c] | 1651 | 0.21[c] |
| **PLA probe section** | 595 | 1590 | 946 | 0.05 |
| **Sandy, silty or clay soils** | | | 1200 – 2800[d] | 0.2 – 2.2[d] |

**Table 1.** Thermal properties of commonly used 3D printed plastics (PLA: polylactic acid, PETG: polyethylene terephthalate glycol), and a section of the printed probe, compared to typical values of sandy, silty or clay soils. Note that the values for PLA and PETG are for objects made out of massive plastic, unlike most 3D-printed objects. [a]Prusa Research (2018, 2020).[b]At ~50 °C (Farah et al., 2016). [c] Rigid.ink (2017). [d]Over the full range of volumetric water content and air-filled porosity (Ochsner et al., 2001).

## 2.2 Fused filament 3D printing

To manufacture this design we used consumer-grade 3D printing technology. The most common consumer-grade 3D printers make use of the 'fused filament fabrication' method, where a computer guided 'extruder' heats up plastic filament and deposits it in the right location to form an object (Chua, Chee Kai; Leong, Kah Fai; Lim, 2003). The extruder can only print lines with a width of the nozzle (most commonly 0.4 mm). The printing is done layer by layer to slowly build up an object in the vertical axis. A limitation of this method of 3D printing is that new layers have to be supported by layers below, which puts a limitation on the shapes that can be printed without adding support material. So-called 'overhangs' are possible, but have a maximum angle of around 50 degrees before the plastic will droop.

For our sensor the plastics PLA and PETG can be used as printing material. These two materials are easily printable on consumer-grade printers, without the need for post-processing or special enclosures. PLA is not recommended for parts exposed to sunlight, or in places where soil temperature exceeds 50 °C, as it is not heat resistant or UV-stable.

The bulk density of printed objects can be substantially lower than the material density, as only the shell is made of solid plastic. In contrast, the internal volume is printed with so called 'infill'. This infill can take on different structures. Generally, the infill of a print is set to a certain percentage of the volume, e.g., an infill of 40% means that 40% of the internal volume consists of plastic and the remaining 60% is air. The bulk density of a printed part can be determined by dividing the final weight of the printed part by the volume the part takes up (as derived from the 3D model). The chosen infill structure will depend on the required properties. In this case we chose for a 'cubic' infill, which will fill the volume with tessellated cubes. This structure will create enclosed pockets of air, which will hinder convection and as such reduce the heat flux through the printed part.

The material out of which the probe is constructed has quite a high heat capacity (Table 1). However, due to the hollow structure of the 3D printed parts and the polyurethane foam core of the probe, the bulk thermal conductivity and heat capacity will be lower than the soil.



A middle section of the probe printed in PLA plastic has an effective infill of 48%; more than half of the volume of the object consisted of air pockets. This results in a heat capacity which is lower than most soils, even when the soil has a high fraction of air-filled pores. The effective thermal conductivity is at least a factor of four lower than dry soils, thus causing a minimal effect of the probe itself on the distribution of temperature in the soil.

### 2.3 Probe assembly

The parts are 3D printed on a Prusa Mk3 printer (Prusa Research, Prague, Czech Republic), using Prusament PLA filament for all segments embedded in the soil and Prusament PETG filament for the top (Prusa Research, 2018, 2020). No post-processing of the printed parts was needed. All segments are glued together using cyanoacrylate adhesive ("superglue"). We used five segments, making the total length of the probe 50 cm, out of which 45 cm has a groove for the fiber optic cable. The remaining 5 cm is smooth.

A fiber optic cable with a diameter of 1.6 mm is routed via the top through the hole at the bottom of the helix. The cable is then coiled around, using the groove as a guide. While coiling the cable it is glued in place using cyanoacrylate glue, which will provide a good bond between the cable and the PLA or PETG plastic. As such a small amount of glue is used, we neglect its thermal properties. When getting near to the end of the spiral, the rest of the cable will have to be routed through the top hole. After this is done the remaining part of the spiral can be glued in place. When the spiral is in place, the bottom cap can

be added to the probe, and the core can be filled with expanding polyurethane foam. This filling will prevent vertical transport of heat or water ingress through the core of the probe. Finally, the top cap can be installed to finish the probe.

### 2.4 Probe and reference sensor installation

The probe was tested at the Speulderbos site in the Netherlands (52°15'N, 5°41'E). The probe was installed near the flux tower at the site, which is located in a plot of ~34 m tall Douglas fir trees (*Pseudotsuga menziesii*). The forest floor is mostly in the shade, except for short periods during sunny days where the light filters through the canopy. The soil in the forest floor has a

1 – 4 cm layer of moss and needles (O-horizon), followed by a dark A-horizon of around 4 cm in depth. This is followed a sandy C-horizon up to at least 40 cm. Due the location being on a sandy hill, the groundwater is many meters deep (Tiktak and Bouten, 1994), and, as such, the soil is always unsaturated.

The soil probe was tested between 15 July 2020 and 30 September 2020. To install the it, a layer of moss and needles was

carefully removed and placed to the side. A hole was pre-drilled using an auger (75 mm diameter), and the probe was inserted into the soil by screwing it in place, leaving the top 3 cm sticking out (Fig 3). After this, some of the sand that was removed with the auger was flushed back in using water, until no more sand flushed down. Some removed moss and needles was carefully placed back around the probe, to restore the previous soil cover.

For reference, four Onset TMCx-HD temperature sensors were placed into the soil at a distance of ~50 cm from the DTS

probe. The sensors were connected to an Onset HOBO 4-Channel External Data Logger (HOBO U12-008). In this setup the temperature sensors have a manufacturer specified accuracy of ±0.25 K. A hole was dug and the sensors were horizontally inserted into the soil at depths of 0, 10, and 30 cm (Fig. 4). The litter layer consisting of moss, twigs and needles has a depth



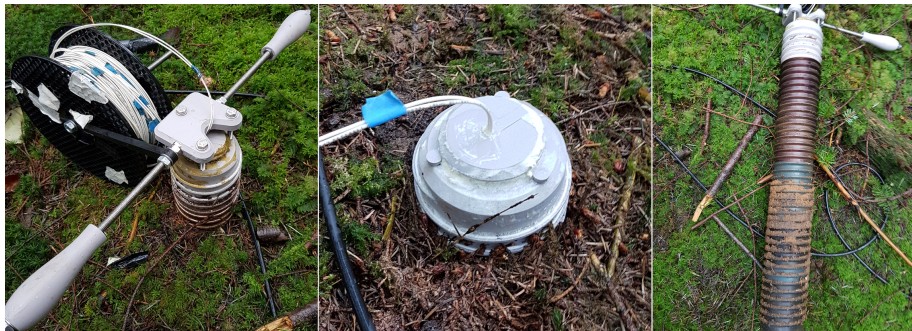

**Figure 3.** Installing the DTS probe using the installation tool (left), the probe during installation (center), and the probe after removal (right).

of varying between 2 and 5 cm. Another temperature sensor was inserted within the litter layer, approximately 2.5 cm above sensor at the soil-litter interface. The depth of this sensor will be represented by a value of -2.5 cm.

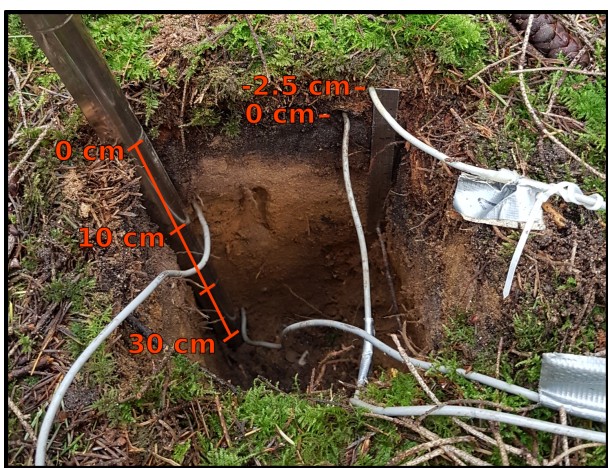

**Figure 4.** Installation of reference sensors. One sensor was placed in the litter layer (2.5 cm above the soil-litter interface), the others at 0, 10 and 30 cm depth relative to the soil-litter interface. Notice the decreasing organic content with depth.

## 2.5 Fiber optic configuration and calibration

For performing the DTS measurements an Ultima-M (Silixa Ltd., Elstree, UK) DTS unit was used. The Ultima-M was housed in a small container on the forest floor. The DTS probe consisted of one long FO cable without splices. From the DTS machine the cable was routed out of the container, through a heated bath, and through a bath at ambient temperature. Both baths were kept mixed using aquarium air pumps with air stones. After the baths the cable was lead to the measurement location under a suspended steel cable, to avoid rodents and to protect the fiber from falling branches. After going through the 3-D printed



probe, the FO cable was routed back under the steel cable, through both baths, and back into the container. Originally the setup was intended to be double-ended; where the fiber is interrogated from both sides as to improve calibration. Due to rodent damage at the container, which occurred near the start of the measurement period, the fiber was only measured in a single-ended configuration. This could cause a small systematic error in areas where the fiber is strained, such as in the tight coil

of the probe. To ensure the accuracy of the measurements, data from the first day (when the fiber was not damaged yet) was calibrated in both single- and double-ended configurations. The difference between the two was insignificant and as such we proceeded with calibrating the DTS data in single-ended configuration during the entire measurement period.

As the DTS measurements will only provide temperature as a function of the *length along the optical fiber*, these coordinates have to be transformed into depth values. This consists of two steps; translating the scale from meters along the fiber to

centimeters along the coil by using the dimensions of the coil (radius, pitch, height), and aligning the soil surface of the probe. The surface can be aligned ('benchmarked') by doing a heat trace experiment, where a specific section of the coil (e.g., part sticking out above the surface) is heated or cooled, and the location along the length of the fiber where there is a sharp temperature spike is noted down. Alternatively, the probe can be aligned by comparing its temperature to the temperature of a reference temperature sensor at a known depth, and minimizing the difference between the two. In this study the final depth

alignment of the probes was performed using the reference sensor at 10 cm. This depth was chosen because small local effects will average out across the soil. To avoid misalignment due to a constant bias between the two sensors, the amplitude of the diurnal temperature oscillation was used for alignment.

## 2.6 Determining soil diffusivity

The thermal diffusivity of a medium can be determined by inverse modeling, that is by using a measured temperature profile

through time. If we assume that the medium is homogeneous, the diffusion of heat through the soil can be described by the following equation:

$$\frac{\partial T}{\partial t} = D\frac{\partial^2 T}{\partial z^2} = \frac{\lambda}{C}\frac{\partial^2 T}{\partial z^2} \tag{1}$$

where $T$ is the soil temperature (K) at a certain depth $z$ (m), $t$ the time (s), $D$ the thermal diffusivity ($m^2$ $s^{-1}$). With only a temperature profile over time and depth, we cannot discern between the thermal conductivity, $\lambda$ (W $m^{-1}$ $K^{-1}$), and the heat

capacity, $C$ (J $m^{-3}$ $K^{-1}$). This would require that soil properties are determined in a lab, or a heat flux plate is installed next to the measured profile. Note that in equation 1 the effects of latent heat fluxes or heat transported by the movement of air or water are neglected (Steele-Dunne et al., 2010).

Since the observations provide information of the soil temperature over depth and time, in principle the 'effective' $D$ could be calculated directly from the discretized version of equation 1. However, attention has to be given to proper estimation of the

second derivative, because small observational errors may lead to a large uncertainty. Here we choose to estimate the diffusivity by fitting a numerical model of Eq. 1 to the measured temperature data, assuming that the diffusivity is constant in time over the period that is studied. We used a (second-order) central finite difference equation (Vuik et al., 2007) to describe the evolution of temperature through time, for a section between two depths. The measured temperatures at the top and bottom of this section




are prescribed, and the temperature in the middle is modeled with an estimate for the diffusivity. By comparing the modeled

temperature to the measured temperature the difference can be minimized, and as a result the apparent diffusivity is determined

$$\frac{\partial T(z)}{\partial t} \approx D \frac{T(z+\Delta z) - 2T(z) + T(z-\Delta z)}{(\Delta z)^2} \tag{2}$$

Due to the large amount of measurement points of the DTS probe, this equation can be used to determine the thermal diffusivity of the soil as a function of depth over the entire vertical profile. This could be expanded upon by incorporating more nearby measurement points for more accuracy, instead of the three points of Eq. 2.

While Eq. 2 has equidistant spacing, this is not necessarily required. For an irregularly spaced sampling, such as with the separate reference probes, the equations can be adjusted (Fornberg, 1988; Taylor, 2016). However, as the reference sensors only measured at four locations in depth, the thermal diffusivity can only be calculated for two (overlapping) sections.

## 3 Results and Discussion

### 3.1 Sample temperature profile

To demonstrate what the data from the probe look like, Fig. 5 shows the evolution of the soil temperature profile over a 24 hour period; 1 August 12:00 until 2 August 12:00.

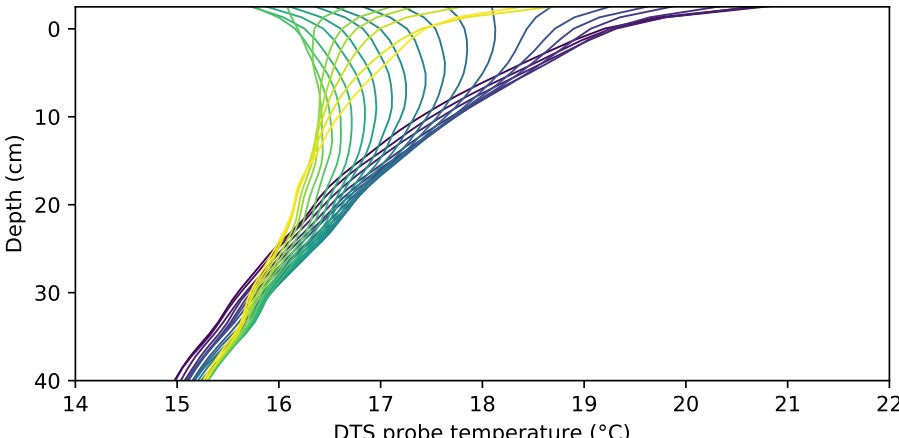

**Figure 5.** Temperature over depth as measured by the DTS probe. Hourly data from noon at 1 August 2020 (dark purple) to noon 2 August 2020 (yellow). The highest and lowest surface temperature occurred at 13:00 and 5:00 respectively.

Starting at midday, the temperature profile is warmest at the top, cooling monotonously towards the deepest measurement at 40 cm depth. Soon after this point in time the soil near the surface starts cooling continuously until early morning. However, the soil below around 30 cm depth continues warming throughout the entire period shown (probably as a result of a long-time





scale, i.e. seasonal, trend in the weather). Note the extremely large gradients near surface. Those gradients are very difficult to measure accurately using traditional temperature sensing. During the night the surface cools down the most, creating a zone 10 – 15 cm below the surface which is warmer than both the soil above and below. This local maximum persists until the soil warms up in the morning and a monotonous profile returns. Due to the very exact vertical spacing and high resolution of the DTS probe, phenomena such as this 'hockey stick' profile can be observed.

The soil temperature deeper down was continuously coldest, as both 1 and 2 August were relatively warm days. A downward heat flux at 40 cm was observed from the start of the measurement period (15 July) until late September.

## 3.2  Comparison with reference sensors

To compare the probe with the reference temperature sensors, the root mean square error between the DTS probe and the reference sensors is computed. However, to reduce any systematic biases between the probe and reference sensors, the data is first
detrended based using a five-day moving average. Systematic biases in the temperature can have their origin in, for example, a slight difference in calibration between the probe and reference sensors. The mean biases between the four reference and the DTS probe ranged between 0.10 and 0.30 K, within the expected uncertainty of the reference sensors and DTS measurements.

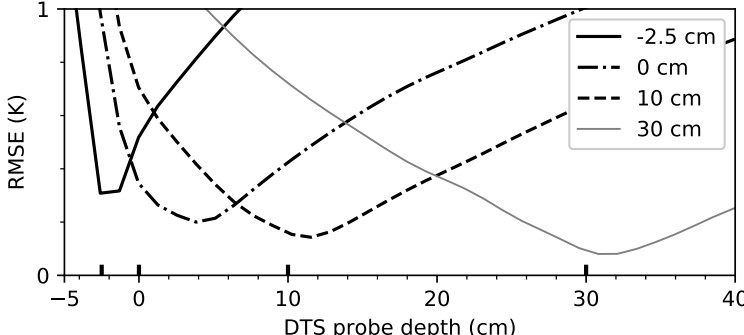

**Figure 6.** Root mean square error (RMSE) of the DTS probe temperature compared to the reference sensors, for the four available depths. The 0 cm depth is the soil-litter interface, and the -2.5cm depth represents the temperature in the moss & litter layer.

Figure 6 shows a good agreement between the DTS probe and the reference sensors, apart from the reference sensor at the soil-litter interface. Even though the probe and reference sensors were placed in relatively close proximity to each other,
variations in the thickness of the litter layer could case discrepancies near the surface. Note that the largest error is also expected near the top where the diurnal amplitude is largest (no scaling of the error is applied).

A second way to compare the probe to the reference sensors is to study how much the diurnal variations in temperature are dampened as depth increases. By comparing the data in this fashion, biases in the absolute temperature are not relevant. Figure 7 shows the temperature variance as a function of depth (after detrending using a 5-day running mean, to only study
the diurnal temperature variation). The variance decreases most strongly in the litter layer, and flattens off deeper into the soil. The reference sensors agree mostly with the DTS probe. As calibration was performed with the 10 cm probe, both agree by





definition. However, the litter and the 30cm sensors seem to be in agreement with the DTS probe as well. The sensor at the soil-litter interface deviates again, just as it did in the previous results. Again, it seems that the deviating reference sensor should be at ~4 cm depth, just as Fig. 6 shows. However, the physical distance between the reference sensors in the litter and the deviating sensor is 2.5 cm, as can be seen from the ruler in in Fig. 4. The source of the deviation is unlikely to be a too-low spatial resolution of the DTS data, as its spatial resolution is 3 cm.

Between the depths of 15 to 40 cm the DTS probe shows an approximately linear decrease in the logarithm of the temperature variance. This exponential dampening is expected when the soil thermal properties do not vary significantly over depth (Moene and van Dam, 2014). That the stronger the slope is, the stronger the dampening. Just above the surface, which is covered by a layer of moss and litter, dampening is strongest. Deeper down the dampening is weaker as there is less organic matter in the soil.

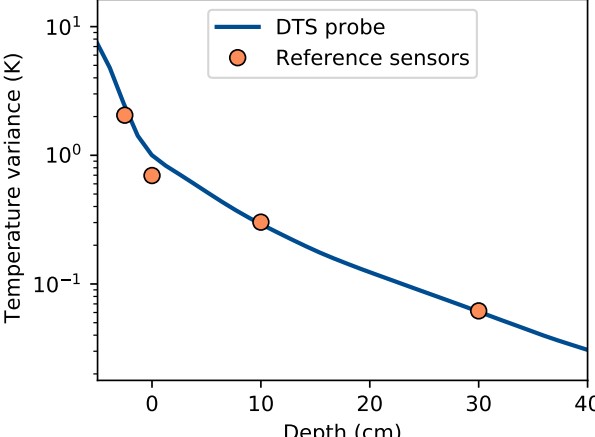

**Figure 7.** Relationship between the temperature variance and depth for DTS probe and reference sensors. Depth values smaller than 0 cm denote parts of the coil possibly sticking out above the surface. Variance was determined on the whole dataset, after detrending using a 5-day running mean.

### 3.3 Thermal diffusivity

The availability of an almost continuous temperature profile by the DTS allows for determination of the soil thermal diffusivity as a function of depth. As there are many data points distributed over the depth, the thermal diffusivity can be estimated in many more intervals compared to standard sensors, as at least 3 measurements of temperature are needed to compute the diffusivity. For the reference sensors only 2 diffusivity values could be computed, using either the sensors at -2.5, 0 and 10 cm, or the sensors at 0, 10 and 30 cm. For the DTS probe data we chose to estimate the diffusivity over increasingly large intervals, from a 2.5 cm wide interval near the surface, to a 10 cm wide interval near the deeper measurement points. This was required as the signal becomes weaker the deeper you go down into the soil, making the uncertainty in the estimate of diffusivity higher.





A difficulty of the separate reference sensors is that any uncertainty or error in the relative depths will directly translate into uncertainty or errors in the diffusivity estimate. If, for example, sensors are slightly further away in reality compared to how they are assumed to be, a higher diffusivity value will be found.

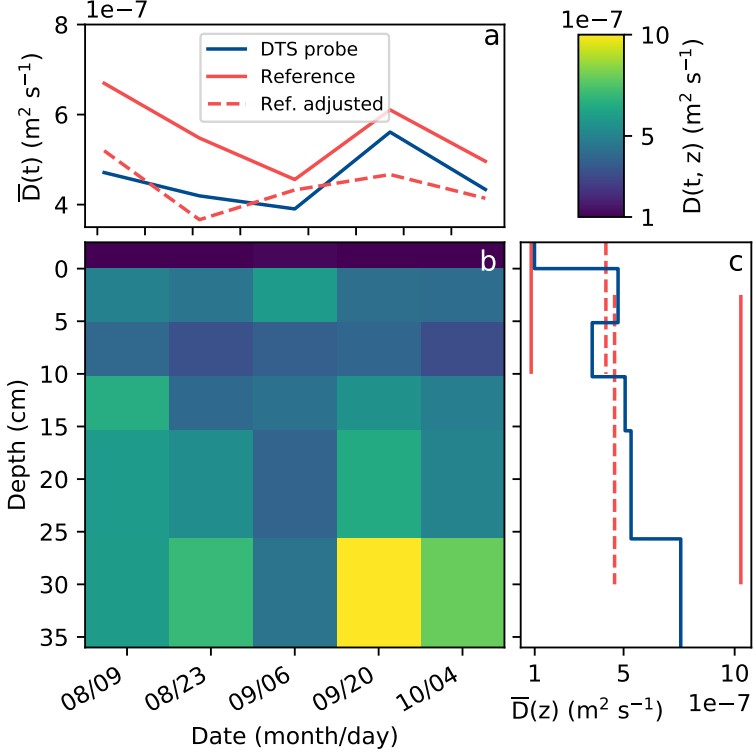

**Figure 8. a:** Mean diffusivity as a function of time, of both the reference sensors and the DTS probe. The red dashed line shows the reference if the second sensor depth is assumed to be at 4 cm depth instead of 0. **b:** The diffusivity as measured by the DTS probe as a function of depth and time. **c:** Mean diffusivity as a function of depth, of both the reference sensors and the DTS probe.

        Figure 8 shows the computed diffusivity values as both a function of time and space. In time the DTS probe and the reference sensors show a very similar pattern in the variations of diffusivity, with just a slight difference in the absolute value. As in Figure
7 we saw that the second reference sensor correlated better with the probe's temperature at 4 cm depth, the diffusivity values of the reference data with this adjusted sensor was calculated. This data shows a pattern in the diffusivity over time that correlated less with the DTS probe data, although the mean error is smaller. The adjusted data shows barely any variation in diffusivity over depth.

        The change in diffusivity over time is to soil moisture; however, for sandy soils this is non-linear with very low moisture
contents (under 0.07 kg kg$^{-1}$, Abu-Hamdeh (2003)). With higher moisture contents the thermal diffusivity of sandy soils is relatively insensitive to moisture, and typically has values around $5 \times 10^{-7}$ m$^2$ s$^{-1}$ (Abu-Hamdeh, 2003; Moene and van Dam, 2014).



Over depth the data of the DTS probe shows a lot more resolution, from the poorly conducting litter layer at the top (dark blue), to higher diffusivity values deeper down (green and yellow). These values of diffusivity vary due to variations in the soil composition and structure, depending on the content of organic matter or, e.g., gravel. Near the surface the reference sensors at the DTS probe agree in the diffusivity, but for the deeper layers the reference sensors estimate a higher value. The value derived using the DTS probe is closer to the expected value for sandy soils. The two methods previously agreed in the temperature variance at the depths of 10 and 30 cm, and as such the deviating sensor at 0 cm would be the most likely cause of the error. Even a slight misalignment such as inserting the sensor at an angle could cause the actual measurement depth to deviate by a centimeter or more.

## 3.4 Outlook

While the DTS-based soil temperature probe does perform well, and could provide more information than conventional sensors, it is important to consider the cost of DTS interrogators. With the high cost associated with these devices, and the many other possible applications for them (Selker et al., 2006; De Jong et al., 2015; Hilgersom et al., 2016; des Tombe et al., 2018; Izett et al., 2019; Heusinkveld et al., 2020), it would not be logical to use them for soil temperature measurements alone. Even so, the probes can be integrated within a network of other DTS measurement of, for example, air temperature or horizontally distributed soil temperature. As long as the DTS interrogator has available measurement length the FO cables of the different setups can be spliced together into a single continuous fiber, and the entire setup can be measured at once.

To make calibration easier and less dependent on calibration baths, two standard soil temperature sensors could be integrated into the probe. This would allow calibration of the probe even if its more fragile FO cable is spliced to a more manageable and rugged FO cable. As a bonus the calibration baths would not be required anymore, which can simplify the setup.

Lastly, the coil does not need to be fully installed into the soil, and can be allowed to stick out into the vegetation (if present). This way, some indication of the temperature profile inside the vegetation can be obtained, which can be used to determine the heat transfer through the vegetation (van der Linden et al., 2022). Do note that, due to the much lower conductivity of air, the coil will have a slow response to temperature changes. Additionally, both incoming and outgoing solar radiation can pose issues, and cause a bias in the measurement (see Schilperoort (2022), Chapter 3).

## 4 Conclusions and recommendations

In this study we presented a design for a DTS-based soil temperature probe, and we tested its performance in the field. The results were in general agreement with the reference sensors, and were able to show more detail than standard sensors are capable of. It was possible to determine the thermal diffusivity of the soil in resolutions down to 2.5 cm, which is many times better than the capabilities of discrete probes. With the higher resolution temperature data, the thermal properties of layers in the soil can be determined at a higher resolution.

Although this study only looked at the accuracy of the temperature measurement, the sensor can be expanded upon by using active heat tracer experiments. This would involve integrating an electrical conductor into the probe, e.g., a metal-tube fiber



285  optic cable, and heating this using its electrical resistance (Bakker et al., 2015; van Ramshorst et al., 2019). If the power is supplied in a pulsed manner, the transient response can be studied to derive the the heat capacity and thermal conductivity over depth (Sayde et al., 2010; Striegl and Loheide II, 2012). Using these properties the soil moisture over depth can also be inferred, e.g., as in Wu et al. (2020). In this case two separate probes can be used, where one measures the undisturbed background soil temperature, and the other measures the thermal properties of the soil. As soil moisture it is an important

290  variable in surface hydrology and land-atmosphere interactions, the continuous monitoring with a combination of vertical and horizontal distributed measurements could provide the best of both; capturing spatial heterogeneity while not sacrificing the accuracy.

**Design availability**

The design files for all parts of the 3D printed probe, including the installation tool, are available on https://www.github.com/

295  BSchilperoort/dts_soil_coil, as well as 3D renders of all parts.

**Data availability**

The processed measurement data is openly available on Zenodo, see Schilperoort and Jiménez-Rodríguez (2023).



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
