# Peer review of "A Distributed Temperature Sensing based soil temperature profiler"

_EGUsphere, 2023_

## Author Comment (AC1)

Thank you for taking the time to review our manuscript. We have replied point by point below (comment in grey, reply in black).

*In this paper, the authors describe a soil temperature probe construction that provides an almost continuous and accurate temperature profile. This probe is built using a 3D printer making it affordable. The concept is welcome as the instrument duplication cost is often a blocking point. However, following scientific deontology, existing probe design should be signaled and some raised or crucial points developed.*

*Indeed, a commercial probe for soil water content and temperature has a very similar design: Campbell SCI "SoilVue10" https://www.campbellsci.com/soilvue10. It is a discrete-level measurement probe; however, the entire sensor is screw-like. Also, a reference to "Soivue10" should be added as this probe exists already even if the employed technique is not the same.*

> Thank you for raising this point. The SoilVue10 probe indeed resembles the DTS probe in its screw-like form. We will add a description of the SoilVue10 probe to the manuscript and will compare its design to the DTS probe (discrete vs. continuous measurements).

*A raised point but not developed is the inhomogeneity of the soil or the soil surface exposition to the sunlight. How these inhomogeneities can affect DTS probe measurements?*

> Inhomogeneities in the soil properties and soil temperature will affect the DTS probe differently compared to traditional measurements. As the path of the fiber optic cable is a helix, the measurement represents a spatial average along part of that helical path, instead of a single point in space. This will cause some (horizontal) spatial averaging. We will add this information to the manuscript.

*Another crucial point needs to be explicitly aborded even if it was not checked more than for 3-month measurements. The soil may be a very aggressive medium, especially for plastic used for 3D impressions. The 3D-printed DTS probe should resist soil humidity, eventual soil acidity, shrinkage, and so on. It may make it difficult, if not impossible, to use 3D-printed probes in some soils. I understand that it would be difficult to develop a point that was not extensively checked but I suggest including a warning for interested readers about it to push them to check their soil compatibility with the planned plastic filament before they invest in a similar project.*

> In our deployments, even in further use in different locations, we have not seen any degradation of the probe. However, to be safe the probe can be manufactured using a more resistant plastic (such as ABS). If soils experience large amounts of shrinkage, the 3D-printed probe is unlikely to perform well. We will add a warning to the readers in the manuscript.

*Finally, It would be very interesting to see authors continue their works and develop a probe as given in the Outlook paragraph and conclusion.*

> Thank you for your interest, we hope that we will be able to do so as well.

*There are some remarks:*

*L-1. Formally speaking, it is not the heat storage that is the component of the energy balance but the heat flow through the soil surface, as the balance is for the energy flow, not the energy storage.*

> We will correct this in the revised manuscript, e.g. to "Storage change of heat in the soil…".

*I think there is a little confusion about the energy flux balance and the soil temperature measurements, as the heat flux measurement is possible using several techniques (usually soil heat flux plates) but is also possible using a soil temperature profile. In the last case, it requires a temperature profile measurement in the soil versus the depth and time. Only surface measurement cannot allow soil heat flux measurement. The goal of the soil temperature measurements needs to be clarified.*

> The skin/surface temperature of the soil is indeed not sufficient to measure the soil energy flux. However, knowledge on the skin temperature is important for modelling the land-atmosphere heat exchange from the perspective of meteorologists.

> I see how the introduction focuses on this too heavily when viewed from a different perspective. We will modify the introduction to clarify the goal(s) of soil temperature measurements.

*L-26 "is strongly heterogeneous due to larger organic matter content" The soil inhomogeneity and organic matter presence are a possibility, not a general characteristic.*

> We will change this to "*can be* strongly heterogeneous…"

*L-70 Unfortunately, this design does not guarantee good soil contact with the sensors in the case of vertisol, such as clayey soil. The same problem arises for the described sensor, and the statement of line 70 must be qualified.*

> We will add a qualifying statement to this line.

*L-96 PLA does not afford prolonged wet soil contact; is PTG resistant enough to soil humidity and, eventually, acid conditions? Was the probe checked for aging once it was installed in the soil?*

> PLA can indeed be more susceptible to degradation in the soil, however under many conditions it will barely degrade[1]. It is only biodegradable under industrial composing conditions. PETG however, is very resistant to weathering[1] from moisture, acidity, etc. The Polyethylene terephthalate (PET) polymer is used for (very acidic) soft drinks for this reason. If a very long lasting probe should be manufactured, ABS plastic is likely the best choice.

> In our deployments we have not observed any effects of aging. However, humidity in PLA plastic can cause it to become brittle.

> [1] https://doi.org/10.1002/gch2.201700048

*L-171 The formula 1 supposes not only that the medium is homogeneous but also that the heat exchange is uniform on its surface (1D heat flow; there is no lateral heat exchange).*

We will add to the manuscript that equation 1 and 2 are based on the assumption of 1D heat flow.

*Figure 6. If I understand well, the temperatures of the reference probe were compared with the temperatures measured by the DTS probe all along the profile measured by the DTS probe. In this case, we have always had the best temperature correspondence, not for the same depth but about 1.5° lower (on DTS) for 10 and 30 cm depth, about 4cm lower on DTS for 0cm depth, and relatively good correspondence for the top of the liter. Is this shift resulting only from the liter thickness? I guess this point was particularly well-checked during reference probe installation.*

The DTS probe and the reference sensors were installed in close proximity, however soil properties such as the litter layer thickness or density can still vary over a small distance. Deeper down the two methods are in high agreement as there will also be horizontal heat transfer which will reduce the impact of local surface inhomogeneities.

We cannot state conclusively if this shift results only from the litter thickness, or if other processes contribute to the shift.

*L-230 The dampening is related to the soil density, not only to the organic matter content. The liter density is much lower than the soil density.*

We will modify this sentence to "Deeper down the dampening is weaker due to the lower organic matter content and higher soil density."

*Figure 8-a "mean" means: averaged on the depth? If not, which depth is compared?*

The "mean diffusivity as function of time" indeed is the diffusivity averaged over depth. We will make this more clear in the revised manuscript.

*L-253 Please do not mix the diffusivity and the conductivity. It is tied, of course, but not the same. The liter is "poorly conductive" but also has a low heat capacity so poor conductivity is not enough to explain the low diffusivity.*

We will change this to "the less diffusive litter layer…"

*L-259. To have an error of 1cm depth measurement with non-null angle insertion on the end of the 50 cm probe (the resulting error is most important on the end of the probe), the angle of insertion should be greater than 11° which is rather unlikely.*

We intended the misaligned sensor to be the reference sensors. Inserting a 5 cm soil temperature probe horizontally into the soil requires an angle of insertion of smaller than 11° to be.

*L-264 It would be interesting to have an estimation of the described probe cost. This is certainly one of its advantages.*

We will add a cost estimation of manufacturing the probe. Do note that the cost of a DTS interrogator of sufficient quality (>50 000 EUR/USD) will far exceed the cost of making one DTS probe. However, the interrogator can be used for other experiments at the same time.

---

## Author Comment (AC2)

Thank you for taking the time to review our manuscript. We have replied point by point below (comment in grey, reply in black).

*In this study, the authors introduce a DTS-based soil temperature probe which allows for the estimation of thermal diffusivity at high spatial resolution. I found the paper to be interesting and well written. I enjoyed the fact that the design of the probe is open-access. I found that the results of the DTS system compared to reference sensors are consistent and promising.*

*I only have a few comments and questions :*

- *The authors claim that " It was possible to determine the thermal diffusivity of the soil in resolutions down to 2.5 cm". Could the authors explain where this "2.5 cm" interval comes from ? What is the spatial sampling of the DTS measurements ? and how do you determine the position/location of the temperature point measurements around the probe ?*

  Please see our answer to the next comment for the question on the resolution.

  The determination of the vertical positions of the temperature measurements is explained in L. 158 – L. 165. We could not study the location of the points on the horizontal plane (e.g., the north or south side of the probe) as we do not have a sufficient spatial resolution to do this.

- *A crucial consideration is the spatial resolution of DTS measurements.*

  *First, I find that the manuscript does not clearly take into account the difference between the sampling and the spatial resolution (10.3390/s20020570; 10.1029/2008WR007052) (what is the performance of the DTS unit here ?)*

  *Then, I wonder how the spatial resolution of measurements affects the results ? The collected data at sample spacing is not truly independent of their adjacent samples. Here, considering the size of the probe, the issue should be addressed.*

  We indeed did not include sufficient information on the spatial resolution of the DTS measurement and the probe in the manuscript.

  The DTS unit used is able to sample at a 25 cm resolution, and has a spatial resolution of ~65cm. The diameter of the fiber groove is 80 mm (Fig. 1c), which leads to a circumference of ~250 mm. Thus we have a vertical *sampling* resolution of 1.0 cm, corresponding to a vertical spatial resolution of 2.6 cm.

  With this vertical spatial resolution of 2.6 cm we are actually *not* able to determine the thermal diffusivity of the soil in resolutions down to 2.5 cm as we stated before, as we need three independent measurements. The actual spatial resolution of diffusivity is thus ~8 cm. We will correct this in the manuscript and add the information on the resolution of the probe to the revised manuscript.

With a higher resolution DTS machine (e.g., the Ultima-S), the vertical spatial resolution of diffusivity can be as low as 5 cm. We sadly did not use this device in this study.

- *L. 237 "For the DTS probe data we chose to estimate the diffusivity over increasingly large intervals, from a 2.5 cm wide interval near the surface, to a 10 cm wide interval near the deeper measurement points". Could you show the results for each interval ? It would be interesting to see the differences. Why did you decide to present the results with a 5 cm interval (Figure 8) ?*

  We chose to aggregate the diffusivity to ever larger intervals for the deeper measurements, as the gradients become smaller over depth. With smaller gradients it becomes more difficult to accurately determine the diffusivity due to measurement uncertainty. To overcome this worse signal-to-noise ratio we chose to aggregate. However, if the diffusivity is determined over a longer interval the spatial resolution can be increased, at the cost of temporal resolution. We will explain this in the revised manuscript.

- *It seems that results are not consistent in the first cm of soils. Could it be due to the spatial resolution of DTS measurements ? The temperature measured near the surface also depends of temperature measurements outside the soil.*

  The near-surface measurements of diffusivity could indeed be slightly influenced by the spatial resolution of the measurements, due to the 2.6 cm spatial resolution of each measurement point. This means that the data point at -2.5 cm will still be influenced by the temperature at around -3.8 cm, which could be above the litter layer. We will add this caveat to the revised manuscript.

*Minor comments :*

- *In streams, some studies already proposed to wrap the FO cable (1016/j.jhydrol.2009.10.033 ; 10.1029/2011WR011227)*
- *In completement of Bakker and des Tombe, you should cite 1029/2020WR028078, as the study includes the estimation of thermal conductivity.*
- *Concerning references, I have the feeling the most references are works of teams from The Netherlands. It would worst strengthen the past literature (https://doi.org/10.1016/bs.agron.2017.11.003)*

Thank you for providing these references. We do have an unintended bias towards works from the Netherlands, and adding the references provided will improve the manuscript.

---

## Author Response (AR2)

**Author Response**

The following under review papers:

van der Linden, S., Kruis, M. T., Hartogensis, O., Moene, A. F., Bosveld, F. C., and van de Wiel, B. J. H.: Heat Transfer through Grass: ADiffusive Approach, Boundary-Layer Meteorology (under review), 2021.

Schilperoort, B., Coenders-Gerrits, M., Jim.nez Rodriguez, C., van der Tol, C., van de Wiel, B., and Savenije, H.: Decoupling of a Douglas fir canopy: a look into the subcanopy with continuous vertical temperature profiles, Biogeosciences Discussions, 2020, 1–25, https://doi.org/10.5194/bg-2020-216, 2020.

Were updated to:

van der Linden, S. J. A., Kruis, M. T., Hartogensis, O. K., Moene, A. F., Bosveld, F. C., and van de Wiel, B. J. H.: Heat Transfer Through Grass: A Diffusive Approach, Boundary-Layer Meteorology, 184, 251–276, https://doi.org/10.1007/s10546-022-00708-7, 2022.

Schilperoort, B., Coenders-Gerrits, M., Jim.nez Rodr.guez, C., van der Tol, C., van de Wiel, B., and Savenije, H.: Decoupling of a Douglas fir canopy: a look into the subcanopy with continuous vertical temperature profiles, Biogeosciences, 17, 6423–6439, https://doi.org/10.5194/bg-17-6423-2020, 2020.

Furthermore, 'over €50000' was changed into 'over €50,000'.